# Vector control strategies in Brazil: a qualitative investigation into community knowledge, attitudes and perceptions following the 2015–2016 Zika virus epidemic

Dani Bancroft  ,[1] Grace M Power  ,[2] Robert T Jones  ,[2] Eduardo Massad,[3,4] Jorge Bernstein Iriart,[5] Raman Preet,[6] John Kinsman,[6] James G Logan[2]

DB and GMP are joint first authors.

For numbered affiliations see end of article.

**Correspondence to**
Dr Robert T Jones;
Robert.Jones@lshtm.ac.uk

## ABSTRACT

**Objective** The World Health Organization declared a Public Health Emergency of International Concern following the rapid emergence of neonatal microcephaly in Brazil during the 2015–2016 Zika virus (ZIKV) epidemic. In response, a national campaign sought to control *Aedes* mosquito populations and reduce ZIKV transmission. Achieving adherence to vector control or mosquito-bite reduction behaviours, including the use of topical mosquito repellents, is challenging. Coproduction of research at the community level is needed to understand and mitigate social determinants of lower engagement with *Aedes* preventive measures, particularly within disempowered groups.

**Design** In 2017, the Zika Preparedness Latin America Network (ZikaPLAN) conducted a qualitative study to understand individual and community level experiences of ZIKV and other mosquito-borne disease outbreaks. Presented here is a thematic analysis of 33 transcripts from community focus groups and semistructured interviews, applying the Health Belief Model (HBM) to elaborate knowledge, attitudes and perceptions of ZIKV and vector control strategies.

**Participants** 120 purposively sampled adults of approximate reproductive age (18–45); 103 women participated in focus groups and 17 men in semistructured interviews.

**Setting** Two sociopolitically and epidemiologically distinct cities in Brazil: Jundiaí (57 km north of São Paolo) and Salvador (Bahia state capital).

**Results** Four key and 12 major themes emerged from the analysis: (1) knowledge and cues to action; (2) attitudes and normative beliefs (perceived threat, barriers, benefits and self-efficacy); (3) behaviour change (household prevention and community participation); and (4) community preferences for novel repellent tools, vector control strategies and ZIKV messaging.

**Conclusions** Common barriers to repellent adherence were accessibility, appearance and effectiveness. A strong case is made for the transferability of the HBM to inform epidemic preparedness for mosquito-borne disease outbreaks at the community level. Nationally, a health campaign targeting men is recommended, in addition to

## Strengths and limitations of this study

► There are limited examples of direct postepidemic engagement and research coproduction with disempowered groups in Brazil, including pregnant women and communities with lower socioeconomic position.

► Focus groups and semistructured interviews provided rich qualitative data on perceptions of vector control strategies and barriers to community engagement with preventive measures during the Zika epidemic.

► A large sample of community members of different ages from two geographically distinct cities in Brazil promoted generalisability of the study outcomes and recommendations.

► A limitation of the focus groups is that participants were asked about their awareness and interest in repellent clothing, and most were not familiar with these as options for personal protection.

► Since interviews took place in 2017, follow-up sessions may have strengthened understanding of how perceptions of *Aedes*-related diseases changed over time, particularly following subsequent outbreaks of chikungunya and yellow fever virus in Brazil.

local mobilisation of funding to strengthen surveillance, risk communication and community engagement.

## BACKGROUND

Zika virus (ZIKV) is a flavivirus primarily transmitted by *Aedes aegypti*, an aggressive day-biting mosquito found in tropical and subtropical climates.[1] Secondary modes of transmission include sexual contact and blood transfusions, as well as vertical transmission in ZIKV-seropositive women.[2 3] Vertical transmission of ZIKV during pregnancy has been associated with devastating developmental consequences in infected offspring, including microcephaly and other neurological

impairments that are collectively recognised as congenital Zika syndrome (CZS).[4–6]

On 11 November 2015, following a significant increase in the number of children born with microcephaly in Northeast Brazil, the Ministry of Health (MoH) declared ZIKV a national emergency.[7] Given the temporal and spatial overlap of microcephaly cases and ZIKV outbreaks, in February 2016, the World Health Organization (WHO) subsequently declared ZIKV a Public Health Emergency of International Concern.[8] By February 2017, Brazil accounted for 65% of the confirmed cases of ZIKV (N=201 821) and 90% of cases of CZS (N=2632) in the Americas.[9]

Population control of *A. aegypti* is the main line of defence against ZIKV transmission.[10] In addition to natural reservoirs, rapid or unplanned urbanisation has contributed to the metropolitan success of this species, which breeds in areas with poor drainage, such as open drains, water tanks and receptacles created by household waste.[11] Negotiating responsibility in relation to maintenance of communal spaces (eg, the individual, community, government or society more broadly) and failure to identify persistent *A. aegypti* or *A. albopictus* cryptic breeding sites hinders adequate vector control.[12] Chronic underfunding and intervention siloes also further undermine efforts to prevent mosquito-borne disease (MBD) outbreaks.[13]

Individual-level mosquito bite-reduction strategies include wearing long-sleeved clothing to create physical barriers, as well as applying topical mosquito repellents.[14] Non-topical strategies include fabric repellent or insecticide sprays.[14 15] However, many repellents do not provide long-lasting protection and often require reapplication.[16] Integration of repellents or insecticides into wearable materials, a method used to treat military clothing in some settings,[17] may instead provide an effective and scalable prevention strategy that is of value to at-risk communities in Brazil.[18]

To reduce sexual transmission of ZIKV, Brazil's MoH promoted condom use and postponement of planned pregnancy during the epidemic.[19] While international guidelines also advocated the relaxation of antiabortion legislation, in Brazil, abortion is only decriminalised for fetal anencephaly (a lethal birth defect), rape or conditions that risk maternal death.[20 21] As a result, abortion was omitted from the MoH protocol on reproduction rights and prenatal, delivery and postpartum care in response to ZIKV.[19] Instead, Brazil's policy strategy emphasised vector control, technology research and development, and assurance of access to healthcare for individuals with long-term sequelae of ZIKV infection.[22]

In November 2016, the WHO declared the end of the ZIKV epidemic.[23] However, as the epidemic waned, development of the most promising vaccine candidates faced challenges in clinical efficacy trials.[24] Since *Aedes* mosquitoes continue to transmit arboviruses worldwide, the epidemic preparedness community remains concerned about the high risk of future outbreaks of ZIKV and other emerging MBDs.[24–27] Brazil's limited success in controlling *Aedes* populations therefore indicates the importance of investigating the social determinants underlying the 2015–2016 ZIKV epidemic.[22 26]

Successful uptake of mosquito-bite preventive strategies is contingent on the broader sociopolitical context, as behaviour change is strongly informed by family, community, cultural, political and economic factors.[13 26 28] The WHO Global Vector Control Response 2017–2030 outlined engagement and mobilisation of communities as one of its four pillars for effective, locally adapted and sustainable vector control.[26] Despite this, during the 2015–2016 ZIKV epidemic, few examples of direct postepidemic engagement or research coproduction with populations at highest risk of adverse health outcomes following ZIKV infection were observed, including with pregnant women and communities experiencing lower socioeconomic position.[29 30] Funding allocated for social research was also markedly lower in comparison to other disciplines.[30] Therefore, to analyse community experiences of ZIKV and vector control strategies in a Brazilian context,[22 31] we consider the application of Rosenstock's Health Belief Model (HBM).[31 32] The HBM is a widely adopted theoretical framework for behaviour change that has been applied to other qualitative studies investigating MBDs.[33 34]

### Aims

This study aims to identify determinants of low adherence to mosquito-bite preventive behaviours by applying the HBM as a conceptual model for community knowledge, attitudes and perceptions towards ZIKV and vector control strategies in two sociopolitically and epidemiologically distinct populations in Brazil: Jundiaí, a municipality of São Paulo (pop. 423 000) and Salvador, the state capital of Bahia (pop. 2.9 million).[35] To best contextualise these drivers, our additional study objectives were to: (1) elaborate household preferences for vector control strategies, particularly with regard to treated clothing; (2) identify perceived barriers to adoption of prevention behaviours; (3) contrast perceptions of ZIKV control with other mosquito-borne arboviruses; (4) compare normative beliefs of pregnancy postponement and abortion to reduce fetal susceptibility to CZS; and (5) map themes against a theoretical framework for behaviour change.

### METHODS

#### Participant recruitment and data collection

From March to August 2017, focus group discussions (FGDs) with adult women of approximate reproductive age (18–49) and semistructured interviews (SSIs) with male partners were conducted in Jundiaí and Salvador. Both cities have cohorts of children living with CZS.[36 37] The interview topic guide comprised 12 questions covering three main areas of enquiry: (1) perceptions and practices of mosquito control, (2) protecting oneself against mosquito bites and (3) knowledge and

perceptions of ZIKV (online supplemental file 1).[38] All sessions were delivered in Brazilian Portuguese, and the source data transcribed and translated into English for analysis.

## Participants

Participants were purposively sampled and consented to participate in the study. The pregnancy status of women was not taken into account and a sociodemographic survey stratified participants by age (18–30 or 31–49 years). In Jundiaí, recruitment took place in outpatient departments at University Hospital, and data collection in both faculty buildings and a non-government organisation (NGO) run community centre. In Salvador, recruitment and data collection took place in two primary care units. In both cities, men were recruited through community stakeholders and interviewed at private residences.

## Patient and public involvement

The principal investigators from Jundiaí and Salvador are native Brazilian speakers familiar with the study setting and context. To ensure the research question was informed by patients' priorities and experiences, the topic guide was developed and pilot tested with research teams local to the study sites. Additionally, 17 in-depth interviews were conducted with health professionals, including Salvador health professionals working in a primary care unit and in private clinics, and community leaders, with three religious leaders from Kardecism, Candomblé (an Afro-Brazilian religion) and an evangelical Christian church. To disseminate results, those who expressed interest and provided consent were invited to attend a follow-up session to discuss initial findings in September 2017.

## Analysis

In total, 33 transcripts were analysed (table 1). Open coding was performed in NVivo (V.12, QSR International). Theme generation followed Braun and Clarke's six phases for thematic analysis.[39] A preliminary coding framework was established from the topic guide. However, coding was mostly inductive, by grouping prevalent response patterns into higher order categories.[40] Major themes were mapped against the constructs in the HBM (figure 1).[31 32] A concept map for themes was developed to gauge whether there was a credible fit with the HBM (figure 2). The 32-item Consolidated Criteria for Reporting Qualitative Research tool was used to ensure all key methodological issues were taken into account (online supplemental file 2).[41]

## RESULTS

A total of 120 individuals participated in the study: 103 women (60 in Jundiaí, 43 in Salvador); and 17 men. Responses to questions on novel repellents were initially coded: effectiveness; affordability; availability; appearance; comfort; protection; risk; and other. Each were mapped against the HBM as: risk (perceived susceptibility); positive responses such as protection (perceived benefits); willingness to adopt (self-efficacy); negative responses for effectiveness, acceptance or accessibility (perceived barriers); and alternative suggestions (preferred criteria). A finalised concept map comprised of 44 minor themes and 12 major themes grouped under four higher order key themes (figure 2; table 2). Definitions are provided in the codebook (online supplemental file 3).

## Knowledge and cues to action

Participants expressed uncertainty around which vectors transmit ZIKV. In Salvador, several participants accurately described the appearance and behaviour of *A. aegypti*. However, the majority of participants did not differentiate the mosquito from other biting insects and some were misinformed. Dengue was the second most commonly identified MBD, although chikungunya and yellow fever were also discussed. Most participants were aware of the impact of ZIKV infection on pregnancy as a distinction from other infectious diseases. However, sexual transmission was poorly understood, and questions from women that disclosed higher levels of education often related to the pathophysiology of ZIKV and unknown sequalae.

*[P1]: So, [microcephaly] sparked people's interest: "Pow, then really, that's the difference between Zika and dengue and H1N1 [influenza]."*

**Salvador-FGD1**

*[P1]: There are 3 different mosquitoes, right?*

**Salvador-FGD2**

*[P2]: [I understood that ZIKV is transmitted] by the host, yes. But not from person to person… This has not been clear to me until today.*

**Salvador-FGD3**

Many women first learnt about ZIKV and were advised to use condoms when accessing maternity services. Often exposure to public health information in broadcast or print media, including pamphlets and posters, was described. Several mentioned learning about ZIKV online via social media, as well as in workplace or higher education settings. Other external cues to action included direct contact with political representatives, NGOs or community volunteers involved with Zika projects. Health agents were described to inspect households and disseminate public health information about *Aedes* and preventive strategies. One key message often recalled was to remove standing water from around the household and spaces shared with neighbours. Participants from four FGDs also recalled a visit from military personnel to promote clearing of communal spaces.

*[P2]: There was a joint effort that the government [made] in the neighbourhood, like this… It was like D-Day against Zika, dengue…*

**Salvador-FGD4**

**Table 1** Summary of interview transcripts provided for analysis

| | Transcript | Words | Participants | Age | | Transcript | Words | Participants | Age |
|---|---|---|---|---|---|---|---|---|---|
| **Female participants** | | | 60 | | | | | 43 | |
| 1 | Jundiaí-FGD-1 | 4338 | | 18–30 | 19 | Salvador-FGD-1 | 14762 | 6 | 31–49 |
| 2 | Jundiaí-FGD-2 | 4399 | | 31–49 | 20 | Salvador-FGD-2 | 3318 | 6 | 18–30 |
| 3 | Jundiaí-FGD-3 | 4067 | | 18–30 | 21 | Salvador-FGD-3 | 16863 | 5 | 31–49 |
| 4 | Jundiaí-FGD-4 | 3409 | | 31–49 | 22 | Salvador-FGD-4 | 10262 | 4 | 18–30 |
| 5 | Jundiaí-FGD-5 | 1691 | | | 23 | Salvador-FGD-5 | 8103 | 5 | 18–30 |
| 6 | Jundiaí-FGD-6 | 4026 | | 31–49 | 24 | Salvador-FGD-6 | 15619 | 5 | 31–49 |
| 7 | Jundiaí-FGD-7 | 1239 | | | 25 | Salvador-FGD-7 | 13138 | 6 | 31–49 |
| 8 | Jundiaí-FGD-8 | 3012 | | 31–49 | 26 | Salvador-FGD-8 | 9256 | 6 | 18–30 |
| 9 | **Jundiaí-FGD-9** | 1860 | | | | | | | |
| **Male participants** | | | 9 | | | | | 8 | |
| 10 | Jundiaí-SSI-1 | 41 | 1 | | 27 | Salvador-SSI-1 | 619 | 1 | |
| 11 | Jundiaí-SSI-2 | 44 | 1 | | 28 | Salvador-SSI-2 | 346 | 1 | |
| 12 | Jundiaí-SSI-3 | 37 | 1 | | 29 | Salvador-SSI-3 | 208 | 1 | |
| 13 | Jundiaí-SSI-4 | 65 | 1 | | 30 | Salvador-SSI-4 | 407 | 1 | |
| 14 | Jundiaí-SSI-5 | 73 | 1 | | 31 | Salvador-SSI-5 | 269 | 1 | |
| 15 | Jundiaí-SSI-6 | 147 | 1 | | 32 | Salvador-SSI-6 | 367 | 1 | |
| 16 | Jundiaí-SSI-7 | 276 | 1 | | 33 | Salvador-SSI-7 | 298 | 1 | |
| 17 | Jundiaí-SSI-8 | 105 | 1 | | 34 | Salvador-SSI-8 | 239 | 1 | |
| 18 | Jundiaí-SSI-9 | 4312 | 1 | 18–30 | | | | | |

A total of 17 focus group discussions (FGD) with 103 women and 16 semistructured interviews (SSI) with 16 men were included in the analysis. Three FGDs and all semistructured interviews were missing sociodemographic data (age). Jundiaí transcripts were missing the breakdown of participants by focus group. Jundiaí FGD-9 was selected for triangulation. Jundiaí SSI-9 was a deviant case excluded from the analysis.

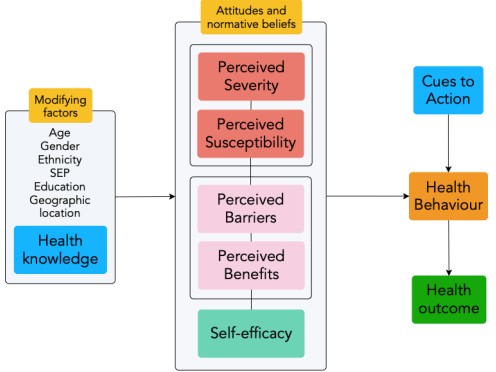

**Figure 1** The Health Belief Model (HBM) adapted from Rosenstock *et al* (1988).[32] The HBM predicts the decision-making process to engage in a new health-seeking behaviour. The individual assesses a perceived threat, potential barriers, benefits and their ability to undergo a behaviour change in response to knowledge, internal or external cues to action.[31]

Visibility of vehicle-mounted thermal spraying/fogging in previous years was recalled by several groups, although most activities were described as having ended. Most agreed that ZIKV messaging had slowed or stopped at the time of their interview, and several participants recalled no community vector control interventions occurring in their neighbourhood at all. Internal cues to action comprised direct or indirect experiences of confirmed/suspected cases of MBDs. In Salvador, more women disclosed having experience of ZIKV infection, whereas in Jundiaí few participants knew someone that had been infected.

*[P1]: I think [during] the outbreak I [became] more attentive… everyone was contracting Zika… Wow! My father had it too, and he had that anxiety thing—if you saw anything, even if it [only] had water in [it] a little while, you'd turn it [upside down].*

**Salvador-FGD4**

### Attitudes and normative beliefs

There was consensus across all groups that pregnant women were most susceptible to ZIKV infection, followed by children, the elderly and those with chronic health conditions. Participants described avoiding travel to areas perceived to present an elevated risk of MBDs, and some understood outbreak seasonality. Several described the

**Figure 2** Concept map of key, major and minor themes for community knowledge, attitudes and perceptions of Zika virus and vector control strategies in Salvador and Jundiaí, Brazil. Four key and 12 major themes were mapped to determine whether they credibly fit constructs for behaviour change outlined in the Health Belief Model.[31 32] The key and major themes are further defined in table 2.

**Table 2** Summary table of definitions for key and major themes.

| Theme | | Definition |
|---|---|---|
| **1.** | **Knowledge and cues to action** | Depth of understanding of ZIKV, MBDs, vector control and key messages identified by participants. Stimuli for a decision-making process that may have led to behaviour change, as recalled at the time of study.[31] |
| 1.1 | Knowledge of MBDs | Participant awareness of MBDs and ZIKV, as well as the community and national response to outbreaks at the time of the study. |
| 1.2 | External cues to action | External stimuli, such as a health campaign, triggered a decision-making process that may have led to a behaviour change. |
| 1.3 | Internal cues to action | Direct and indirect experiences of confirmed or suspected cases of MBDs triggered a decision-making process that may have led to a behaviour change. |
| **2.** | **Attitudes and normative beliefs** | Personal attitudes are internal assessments of knowledge and cues to action for MBD preventive behaviours. Normative beliefs may inform personal attitudes according to how others perceive the behaviour in a social setting, such as the community. |
| 2.1 | Perceived susceptibility | A subjective assessment of the risk of ZIKV infection or a CZS pregnancy and the first component of perceived threat.[31] |
| 2.2 | Perceived severity | A subjective assessment of the severity of ZIKV symptoms and CZS and the second component of perceived threat.[31] |
| 2.3 | Perceived barriers | An individual's assessment of the barriers to uptake of ZIKV preventive behaviours for sexual transmission, mosquito bite-reduction and vector control. |
| 2.4 | Perceived benefits and self-efficacy | An individual's perception of the benefits of novel repellent technologies and their ability to successfully undergo a behaviour change by adopting preventive strategies. |
| **3.** | **Behaviour change** | Behaviours either attributed to the ZIKV epidemic, are pre-existing practices against MBDs (no change), or no preventive measures were taken. |
| **3.1** | Household level | Practices to prevent mosquitoes from breeding and exposure to mosquito bites at the household level. |
| **3.2** | Community participation | Engaging with others in the community; participants describe activities for collective action for vector control. |
| **4.** | **Community preferences** | Expressed needs and preferences for mosquito bite-reduction strategies, coordination of vector control and ZIKV messaging, including questions. |
| 4.1 | Novel repellents | Preferred criteria for novel topical mosquito repellents, repellent-impregnated clothing or other wearables designed to prevent mosquito bites. |
| 4.2 | Vector control strategy | Preferred activities for mosquito population control, including surveillance. |
| 4.3 | ZIKV messaging | Preferred content, source and format for delivery of ZIKV risk communication and community engagement. |

CZS, congenital Zika syndrome; MBD, mosquito-borne disease; ZIKV, Zika virus.

belief that infection by one MBD increased their susceptibility to others, although there was a lot of uncertainty and misinformation around ZIKV case confirmation. The spread of misinformation was a concern to participants, and several misinterpreted or described feeling unable to trust public information about the origin of the virus.

*[P2]: In my opinion, I knew that Dengue and Zika is the same thing… I think that's evolution from one disease to another.*

**Salvador-FGD6**

Living in an area of perceived low risk was often described to diminish participants interest in adopting preventive measures ("*It's only worrisome when there's an epidemic,*" **Jundiaí-FGD1**). However, there was less consensus between focus groups regarding where population density of *A. aegypti* vectors was highest, and several participants described the mosquito as absent from their neighbourhood altogether. Perceived severity of ZIKV infection also varied considerably. Some likened ZIKV symptoms to mild influenza, although women perceived there to be a higher threat from ZIKV than men. Some participants recalled inflammation of the joints and fatigue as symptoms that required extended recovery, and a few described the risk of death as a potential consequence of ZIKV infection.

*[P1]: It caused a panic, right? Many women gave up being mothers, or they delayed, right? Fear of disease.*

 Bancroft D, *et al. BMJ Open* 2022;**12**:e050991. doi:10.1136/bmjopen-2021-050991

*[P2]: In fact, all the [mosquito-borne] diseases mentioned are worrisome, right? Even the flu is scary.*

**Salvador-FGD8**

Some participants also described differences in the appearance of bites from mosquitoes carrying ZIKV. Several likened the experience to an allergic reaction, which is perhaps a perception of maculopapular rash linked to ZIKV infection.[42] Several women demonstrated higher awareness of ZIKV sequelae from work or study in healthcare, or volunteering with local ZIKV projects. Although collective awareness was described to have peaked and waned, several participants commented on the visibility of families caring for a child with CZS in broadcast media, and they believed a greater disease burden was in more deprived communities.

*…usually the people most affected [by CZS] are low-level people right…people who have poor conditions, who live in more inappropriate places.*

**Jundiaí-Male-SSI-8**

Several participants disclosed they would be willing to access abortion services to reduce risk of having a child with CZS or having previously terminated a pregnancy. However, perceptions of rights to terminate a pregnancy were influenced by strong social norms and religious beliefs, and there was often reluctance to disclose or elaborate on personal attitudes due to its criminalisation. Some conceded community attitudes and norms towards abortion were more nuanced given perceptions of lower quality-of-life due to severe disability associated with CZS. However, for one focus group, partial legislation of abortion in the case of microcephaly was criticised as inadequate and perpetuating discrimination.

*…I think it depends on where she congregates because religion weighs in a lot… She will not do it because of religion, and if she dares [abort], she will not be accepted.*

**Jundiaí-FGD4**

*[P1]: Anencephaly in cases of problems was allowed because it makes life unfeasible, but microcephaly does not… So, you're just going to admit normal kids? It'd be a way of sanitizing the population…*

**Salvador-FGD3**

Women aged 18–30 were more supportive of the right to abort, as were participants that disclosed as working in healthcare or having accessed higher education. Despite adequate levels of perceived threat from ZIKV and recognition of potential benefits of a behaviour, participants described many barriers to reproductive health decision-making. There was frustration around the burdens of preventing ZIKV and caring for children with CZS falling on women. Discordant attitudes towards abortion between pregnant women and male partners were also discussed. For example, women reported diminished self-efficacy to negotiate condom use with an intimate partner

during the epidemic, often attributed to the stability of the relationship or harmful gender norms.

*[P2]: We've already talked about machismo, right? I've heard of a husband dropping his wife and saying "No, if you do not [abort], I'll let you go," because she already knew she had a microcephaly [baby].*

…

*[P3]: Yes, but the question of the condom? OK! One part would accept, but this question of non-penetrative sex for six months? No!*

**Salvador-FGD5**

With regards to mosquito-bite prevention, for several participants, skin allergies were also a barrier to the use of topical repellents for personal protection. While this motivated some to consider investing in alternative brands or non-chemical bite-reduction strategies, there was broadly low participant awareness of novel repellent tools such as clothing. While participants were relatively unfamiliar with repellent treated clothing, some recognised the benefit of these items for protecting children and pregnant women, although overheating during pregnancy was a concern. However, in both cities, repellents were described as less accessible for individuals of lower socioeconomic position. Owning a single item was not perceived to provide sufficient protection, yet buying 'a whole wardrobe' would be a significant investment. Interest was also strongly affected by their appearance in the community, including negative perceptions of the association between MBDs and social deprivation.

*[P4]: It's just one more expense, right?*

*[ALL]: Yes!*

*[P5]: It would probably be very expensive. Because it would be, say, for the elite.*

**Salvador-FGD5**

*Maybe he is bothered about having to wear clothes that would be, in this case, also an indicator of poverty, right?*

**Male-Salvador-SSI-4**

Overall, during interviews there was a positive reception to adoption of novel repellent tools. However, similarity was observed between shorter responses and interviewer prompts, and men often expressed disinterest. Scepticism around long-term effectiveness of repellent clothing was also observed, including concern for areas of skin left exposed.

*[P4]: …an entire population can't be protected that way. In particularly endemic regions and for high-risk group like babies or pregnant [women] it works, but it's not good for you to dress a whole neighbourhood with the same clothes!*

**Salvador-FGD4**

At the community level, contextual factors were often linked to MBD outbreaks, such as inadequate coverage of urban planning like sanitation services. In Salvador, the

former administration was criticised for poor management of the ZIKV epidemic, including the cost of testing, financial support for families with CZS children and an over-reliance on mass-media campaigns. Surveillance teams were often perceived as undermotivated or not being trusted to adequately search for cryptic breeding sites. Some participants also described health agents refusing to enter all households in a community, attributed to either concerns around neighbourhood violence or inadequate upstream coordination of vector control efforts.

*[P1]: Where are the community agents themselves? I'm not talking about treatment, I'm talking about preventive measures. Community agents are not effective by municipal power…it's a type of unstable work, you know? There are months without receiving [them].*

**Salvador-FGD3**

*There is a lot of suspicion…total distrust in the [Zika] project… The resistance with men is great.*

**Jundiaí-FGD4**

### Behaviour change

The most frequent vector control strategy described by participants at the household level was preventing water stagnating by recycling, using sand, covering open receptacles and applying detergents or treatments to bodies of water. Bite-reduction strategies included physical barriers: fans, air-conditioning, bed-nets, window screens and long clothing. Several described using plug-in appliances or burning coils to repel mosquitoes with increased frequency during the epidemic. Electric-shock devices to kill adult mosquitoes were also popular. Some participants, particularly pregnant women, avoided travel to certain areas or during times when mosquitoes were believed to be most active. Women in every focus group described knowing someone in their social circle that delayed pregnancy to mitigate the risk of CZS.

*I have two sisters-in-law who wanted to get pregnant, but because of the epidemic they were afraid and postponed it.*

**Jundiaí-FGD3**

Community participation comprised reporting mosquito breeding sites to public health authorities, which was frequently discussed in Jundiaí. Several women described generally observing and encouraging behaviour change in others, including the use of repellents and general maintenance of potential *Aedes* breeding sites.

*[P1]: …it's not just the authorities, everyone has to do their part…to be able to openly reach the neighbour and say, "Oh, look at your bottles [they're] full of water, focus!"*

**Salvador-FGD8**

Although some participants described skin irritation from topical repellents, only one participant recalled women avoiding chemical repellents during pregnancy due to safety concerns. Methods for mixing plant-based oils or alcohol with chemical formulations and sunscreen were described to soothe and prevent bites from becoming infected. Doing so was also described to mask the smell of repellent products and reduce the cost of repurchase.

*[In] Bahia, the desperation is greater than here, and pregnant women are afraid to use any product and use homemade products [instead]…*

**Jundiaí-FGD2**

### Community preferences

Subsidy of contraceptives and repellents were suggested for lower income or high-risk groups during outbreaks. Alternatively, it was recommended that they are freely distributed by local health clinics, NGOs or Brazil's national social welfare programme, Bolsa Família.

*[P4]: The government should give repellent to the people since you have this yellow fever outbreak. Make a campaign. The same people who have family-grants should be entitled…*

**Salvador-FGD6**

When asked what participants thought of treated clothing, repellent school uniforms to reduce children's risk of MBDs and adult sleepwear to mitigate discomfort from bednets or topical repellents were of interest. Microencapsulation of repellents in wearable plastics were also suggested by some, such as bracelets. Generally, participants expressed interest in clothing items if they were affordable, aligned with local preferences in fashion (eg, fabric quality, design) and the smell of repellent product could not be easily identified. However, the ability to renew the effectiveness of existing items was also important.

*[P1]: …you would have to change your wardrobe to buy only mosquito repellent clothes. It would be [a] more effective process [if] you make your clothes have this substance.*

*[P2]: It makes more sense. Like a lotion.*

*[P1]: A lotion that you put on to do laundry…*

*[P3]: Yeah, like a fabric softener.*

**Salvador-FGD4**

For vector control, often improvements in municipal service coordination was expressed as a priority need, citing open drains or infrequent collection of household waste. One focus group was interested in reintroducing thermal spraying of insecticides. Another explored the idea of financing the coordination of neighbourhood associations to mobilise the community, including financial compensation of volunteers.

*[P1]: How are we going to complain about our problems? We do not have a person who can get there and settle for us. If we make a petition, everybody in the neighbourhood will sign, but who will take it? …our neighbourhood is abandoned, we have no association…*

*[P2]: I think every neighbourhood should have [an] association].*

 Bancroft D, *et al. BMJ Open* 2022;**12**:e050991. doi:10.1136/bmjopen-2021-050991

*[P1]: [The former volunteer] did everything for us there. My street was clean, everything was clean. There should be someone to count, take care, understand?*

*[P3]: If she's doing it, she has to get something too…*

*[P4]: But the staff thinks the person [must] work for free.*

**Salvador-FGD6**

There was disagreement regarding the saturation of ZIKV messaging during public health campaigns. The majority of women expressed feeling underequipped with the practical knowledge for prevention. Whilst a few asserted messages were overly technical, others did not feel they provided sufficient detail to implement vector control strategies at the household level. Preference was therefore placed on sustained delivery of messages between outbreaks, via social media or print materials in public spaces. A media campaign that targets men was suggested as one approach to escalate perceptions of the health risks and consequences for intimate partners due to sexual transmission of ZIKV. A sexual and reproductive health-focused curriculum for schools was described as another point of delivery to improve community engagement with messaging. Health promotion materials to facilitate community events were also suggested to amplify the effect of annual awareness campaigns like 'World Dengue Day'.

*[P4]: If it's not in the extreme, [messaging] will not work. It's like cigarette campaigns.*

**Salvador-FGD5**

*No, it's not a lack of information, it's education…it has to start very early with sex education. Because human beings only change their habits when something very serious happens. I think information alone does not [do it].*

**Jundiaí-FGD3**

## DISCUSSION

In the outbreak beginning 2015, Brazil experienced more cases of ZIKV than any other country. Its MoH responded with a policy strategy focused on vector control, improved healthcare access, and technology and research development.[43] However, it has been argued that these policies failed to reach those most vulnerable to the virus.[20 44] The northeast of Brazil was particularly hard hit, as a region with some of the lowest Human Development Indices (HDI) in the country.[37 45] In comparison, in 2017, Jundiaí was ranked as having the 11th highest HDI of 5564 municipalities in Brazil.[36] Individuals from communities in Salvador and Jundiaí were invited to provide their knowledge and perceptions of ZIKV and MBD control for this investigation.

### Community awareness of mosquito-borne diseases
The sessions revealed that participant understanding of their susceptibility to infection was a key influence on their decision-making to engage in health protection measures.

Direct or indirect experience of ZIKV and dengue was a common internal cue to action in Salvador, a city with a long history of MBD outbreaks,[46] which is consistent with previous findings.[27 37] However, participants frequently believed that ZIKV-carrying *Aedes* mosquitoes were absent in their local area, and perceptions varied as to where in Brazil the prevalence of MBDs was greatest. At the time of the study, a national yellow fever vaccination campaign was communicating outbreaks in non-human primates, and some participants discussed fearing reports of its urbanisation.[23 47] Participants describing a potential relationship between ZIKV and other MBDs was not unwarranted, as arboviruses transmitted by *Aedes* tend to cluster.[13] Sequential arboviral infection is also still poorly understood,[45] with some studies suggesting limited cross-immunity following dengue virus infection.[48–50]

The majority of women interviewed were unaware of the risk of ZIKV transmission from unprotected sex. This is consistent with findings from other studies on ZIKV risk communication,[33] including in Colombia.[51] Since interviews were conducted towards the end of the outbreak, this suggests there was a missed opportunity to prevent at least some of the spread of ZIKV. Although the ultimate importance of sexual transmission may be small compared with that of mosquito-borne transmission,[52] the public should receive clear messaging around the relative contributions of mosquito-borne, vertical, sexual and bloodborne transmission, to enable individuals to make informed choices about adopting preventive measures.

### Social determinants of Zika virus and congenital Zika syndrome
There was strong disagreement around the criminalisation of abortion, which has been dismissed as a paternalistic policy that is inconsistent with MoH advice to avoid or delay pregnancy in ZIKV endemic areas.[20 21 53] The sense that ZIKV has been emasculated, where the responsibility to prevent sexual transmission has fallen to women, has also been described in other studies.[53–57] Despite being strongly advocated by international multi-lateral agencies and Brazilian legislators,[20 21] important questions remain outstanding on reproductive health rights for ZIKV sero-positive individuals.[58 59]

MBDs, including ZIKV, predominantly affect individuals in socioeconomically deprived areas.[29 60] Inadequate access to clean water, sanitation and other infrastructural deficits allow mosquito populations to thrive.[26] In addition, individuals in these communities may also be less able to afford tools for personal protection and have poorer access to good quality healthcare.[45 61 62] In our focus groups, the perceived severity of ZIKV was most often framed through the lens of disadvantage: the availability and affordability of amniocentesis or ZIKV testing; female agency to negotiate abstinence or long-term condom use with their male partners; access and acceptance of contraceptives to delay pregnancy or abortion; and uncertainty around a financial and social support

network to care for children with CZS. These themes were consistent with other study findings.[57 61–63]

### Personal protection strategies

Topical repellents are uncomfortable for some users, and may not be seen as long-term solutions for preventing mosquito bites.[15 64] The pay-off for repeat application of repellents may also be less certain for ZIKV than other MBDs, where the onset of symptoms and potential consequences of infection is comparatively short.[65] Novel, nontopical repellent technologies are not yet widely known or understood, and perceived safety of synthetic repellents was anticipated to be a key barrier to their adoption, as seen in other qualitative studies.[34 51] Instead, the key barriers identified in this study were the effectiveness and accessibility of novel repellent tools like clothing.

In Salvador, it was also important that repellent clothing was not perceived to be a 'uniform' associated with low-socioeconomic position, while in Jundiaí, participants discussed the need for clothing designs to reflect local preferences in fashion. The concept of repellent school uniforms to protect school-going children from MBDs was well received and has demonstrated strong potential in a cluster randomised-controlled trial in Thailand.[17] Participants also expressed an interest in being able to renew the repellent effect of clothing to overcome barriers like affordability and durability, negating the need for replacements. For example, using sprays to reapply repellents to clothing was perceived as more feasible option to clothes treated prior to purchase. Some also acknowledged the attractiveness of formulated washes for ease of application, and incorporation of perfumes to mask repellent smell.

### Vector control strategies

Mosquito prevention at the household level was often perceived to be a burden. However, many participants described removal or treatment of potential mosquito-breeding sites as being incorporated into daily routines. Despite this, several individuals expressed their personal control beliefs for vector control were fatigued when neighbours did not also do their part. Abandoned buildings or communal spaces 'contaminating' maintained areas contributed to some participants' sense of futility; even if they were well informed, a public health challenge as prevalent as *Aedes* was not something the community could 'combat' alone.

Minor themes of blame, mistrust and responsibility were also frequently allocated upstream, especially in Salvador. Reporting mosquito-breeding sites in communal areas in more deprived neighbourhoods to the City Hall was deemed unlikely to result in change due to broader inadequacies in local urban planning. Some participants also expressed frustration due to a lack of consistent or thorough household inspections by surveillance teams, confusion around the different stakeholders involved during follow-up visits, or a need for clarification of ZIKV key messages. Often, this was attributed to chronic underinvestment in vector control, a common theme in other studies in South America, where both men and women have expressed a need for intensification of government support.[51 54 56]

### Community engagement related to Zika virus prevention

Freire posits that structural inequalities in Brazil creates a loss of agency,[66] which in the context of the ZIKV epidemic, likely constrained self-efficacy for behaviour change.[44] A systems model for *Aedes* vector control also argues that the pathway between collective awareness, collective action, community attitudes and normative beliefs is simply too long for effective control of MBD outbreaks.[28] The opportunity to communicate barriers in a more timely manner would improve collective awareness, as well as political will for local authorities to act.[13] Carvalho *et al* proposed one solution could be investing in improved frequency of household visits from community health workers (CHWs) under the Family Health Strategy,[28] which covers 66.5% of Brazil's population.[67] Although task-shifting of CHW responsibilities to include ZIKV case reporting was possible during the epidemic, their catchment area excluded middle-income or high-income neighbourhoods,[68] like Jundiaí.

Instead, a community-participation model is proposed. Grassroots approaches, such as neighbourhood associations, may serve as a more trusted setting for community engagement during infectious disease outbreaks.[13 69] For example, in a meta-analysis on uptake of novel repellent technologies, participatory models were found most effective at improving self-efficacy,[70] as well as promoting a sense of community responsibility.[71] Financing mechanisms to decentralise and triage risk communication and vector control at the community level may also mitigate the marginalisation of individuals in more deprived settings, largely caused by top-down approaches in health promotion.[66]

### Limitations

Some participants were not familiar with questions raised on novel repellents in the topic guide. Additionally, the differentiation between prevention measures for ZIKV may not have always been clearly understood. Interview prompts, such as preferences for novel repellents, may have therefore enabled acquiescence response bias.[40] When focus groups discussed more contentious topics, such as abortion, personal attitudes may have also been conflated with social norms if some women felt unable to disclose disagreement with the majority.[72] Although facilitators were able to detect non-verbal cues for each, subtext may have been lost during analysis. To mitigate this, an independent translation service was used to verify the credibility of transcript excerpts, and preliminary findings were discussed with principal investigators for triangulation. Additional data were not collected on participants, such as data on socioeconomic position, which along with missing data on age for some Jundiaí focus groups could have provided an interesting overview of the participants

in this study. The selection of the HBM as a conceptual framework is also necessarily limited,[73] particularly given the scope of themes raised in the topic guide and context-specific challenges reported by other researchers during the 2015–2016 ZIKV epidemic.[30] Nonetheless, the HBM still permitted a relatively deep analysis of individual-level factors, despite disagreement in the literature over the order in which the framework's components may lead to behaviour change.[73] The literature was thus consulted post-analysis for transferability of findings.

## Recommendations

This investigation recommends that national authorities provide effective repellent tools to families entitled to social welfare in settings where MBD outbreaks are regular occurrences, and during outbreaks extend this provision to include high risk groups. Capacity-building of MBD surveillance teams is also recommended to strengthen multilevel governance and reduce gaps in the frequency of interventions designed to prevent infectious disease transmission, such as household inspections. A degree of data saturation for preferred criteria of novel repellents in this study lends weight to the finding there was an unmet need for alternative personal protective tools to topical repellents.

The WHO Global Vector Control Response advises cross-disciplinary community engagement to improve context-sensitive messaging and reduce barriers to uptake of MBD preventive strategies.[26] Designing a mass-media campaign that targets men could improve awareness of ZIKV sexual transmission and emphasise the importance of protecting the health of their female intimate partners. Financing participatory models for community engagement would also demonstrate a firm commitment to translating politicised slogans into an effective, bottom-up control strategy for *Aedes*-related MBDs.

It is worth noting our recommendations are also pertinent to the response to the SARS-CoV-2 pandemic. At the time of writing, Brazil also had among the highest numbers of confirmed COVID-19 cases in the world, particularly in the North, and its MoH was criticised for not developing a national plan to combat the disease.[74] In light of this, further focus group studies, or design of a Likert scale-based survey that operationalises the HBM during data collection,[31] may also prove fruitful for understanding how perceived severity and susceptibility to MBDs has changed in Salvador and Jundiaí, particularly following outbreaks of chikungunya and yellow fever virus.[23 75]

## CONCLUSION

This study makes a strong case for the value of qualitative investigations and transferability of the HBM to inform bottom-up approaches in health protection. Since the initial outbreak in Brazil in 2015, the fall of the perceived threat from ZIKV, normalisation of CZS symptoms in affected children, and the poorly understood relationship to other arboviruses transmitted by *Aedes* has weakened community self-efficacy and perceptions of the government response. Participant awareness of sexual transmission of ZIKV was low and several focus groups discussed an unmet need for a health campaign that targeted men. Significant barriers were also discussed around the affordability of mosquito-bite prevention strategies, such as topical repellents and novel tools for personal protection, including their perception as a potential marker of socioeconomic position. Household behaviours to control mosquitoes were also often fatigued by a lack of cooperation and coordination at the community and municipal levels. It is therefore argued that the historical failure to control *Aedes* outbreaks in Brazil lies in placing too much responsibility on the individual, particularly women. By investing in evidence-based epidemic preparedness, and by stimulating a sense of community agency to tackle vector breeding sites, Brazil may indeed be better placed to 'beat' the *Aedes* mosquito.

**Author affiliations**
[1]Department of Public Health, Environments and Society, London School of Hygiene and Tropical Medicine, London, UK
[2]Department of Disease Control, London School of Hygiene & Tropical Medicine, London, UK
[3]School of Medicine, University of São Paulo, São Paulo, SP, Brazil
[4]School of Applied Mathematics, Fundação Getulio Vargas, Rio de Janeiro, RJ, Brazil
[5]Institute of Collective Health, Federal University of Bahia, Salvador, BA, Brazil
[6]Department of Epidemiology and Global Health, Umeå University, Umeå, Sweden

**Acknowledgements** We would like to thank all study participants, and the ZikaPLAN research teams in Salvador and Jundiaí: Ana Maria Rico, Greice Bezerra Viana, Fernanda Macedo da Silva Lima, Mônica Manir, Tania Boccia and Vera Lucia Zaher-Rutherford. We would also like to thank Alexandra Levitas for her support during analysis.

**Contributors** JGL conceived the study. JBI and EM led data collection in Salvador and Jundiaí, coordinated by JK. DB led the analysis and the University College London Digital Media service was used to translate select excerpts of Brazilian transcripts for verification against the translations made by EM. GMP and RTJ performed triangulation of coding. DB, GMP, RP and RTJ authored the manuscript for publication. DB and GMP are joint first authors. All authors read and approved the final manuscript. RTJ is the study guarantor.

**Funding** This study was financed by the European Union's Horizon 2020 research and innovation programme, awarded to the Zika Preparedness Latin American Network (ZikaPLAN) under Grant Agreement No. 734584. The Department of Disease Control, Faculty of Infectious Diseases, London School of Hygiene & Tropical Medicine, provided funding to support publication.

**Competing interests** None declared.

**Patient consent for publication** Not applicable.

**Ethics approval** Approval for the study in both Jundiaí and Salvador was granted by the Jundiaí School of Medicine Ethical Review Board in January 2017 (REF: 1.875.618). For analysis, approval was granted by the MSc Research Ethics Committee at the London School of Hygiene & Tropical Medicine in July 2020 (REF: 21978).

**Provenance and peer review** Not commissioned; externally peer reviewed.

**Data availability statement** Data are available in a public, open access repository. All data relevant to the study are included in the article or uploaded as supplementary information. The topic guide, codebook and COREQ checklist supporting the conclusions of this article are provided as supplementary files. The consent form and topic guide are also available at the London School of Hygiene & Tropical Medicine (LSHTM) Data Compass repository (https://doi.org/10.17037/DATA.00002097). The transcripts of focus groups and semistructured

interviews supporting the conclusions of this article cannot be made available for confidentiality reasons.

**ORCID iDs**
Dani Bancroft http://orcid.org/0000-0002-5846-5818
Grace M Power http://orcid.org/0000-0002-5702-7728
Robert T Jones http://orcid.org/0000-0001-6421-0881

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
