## [Reviewer comments · BMJ Open]

ARTICLE DETAILS

TITLE (PROVISIONAL)	Vector control strategies in Brazil: A qualitative investigation into community knowledge, attitudes and perceptions following the 2015–16 Zika virus epidemic
AUTHORS	Bancroft, Dani; Power, Grace; Jones, Robert; Massad, Eduardo; Iriat, Jorge Bernstein; Preet, Raman; Kinsman, J; Logan, James

VERSION 1 – REVIEW

REVIEWER	Leontsini, Elli Johns Hopkins University Bloomberg School of Public Health
REVIEW RETURNED	22-Apr-2021

GENERAL COMMENTS	1. The field guide provided shows insufficient differentiation among three types of measures: to control the immature vector stages, to control the adult mosquitoes, and to repel their bites. The responses therefore are equally as vague as the question asked. Please acknowledge as a study limitation.2. Unclear as to why so much emphasis was given to bite prevention – the hardest of all, for a swift day biter that goes unnoticed. Perhaps the authors can provide some justification and acknowledge this as limitation.3. Unclear as to what the novel repellents were. There is a mention of permethrin treated clothing without any information on whether it was available for purchase and use. Inferred from the participants' responses is that this type of clothing was not available for use at the time of data collection. Participants' answers therefore were in no way based on experience but in speculation. Please provide any additional information on this and add it to the study limitations.4. Figure 2 is unreadable as is. There is no room to explain what each concept means. Please cut out some of the minor themes to be able to explain the major ones better.5. Intro, methods and results would benefit from improved writing skills. Senior authors, please pull your weight and bring the writing quality up to that of the discussion section.6. There were so many good themes that emerged but they are over-synthesized in a couple of generic headings coming from HBM rather than from the data. Please use more headings that reveal the theme below the heading. The data provide little on core HBM determinants (risk perception) and a lot more on barriers of use. A socioecological model, or a behavioral model might be a better underlying model to use.7. Table 1. Units of analysis are not understandable by the reader – say transcripts instead. Add number of participants for each FG. Mark on the Table the gender for each unit. What criteria were used to put certain participants to the same FG? Why so many FGs? Participants were purposefully sampled – please state with what purpose, i.e., the selection criteria. Do we know anything about their
--

	profession or SES? Does Table 1 include health professionals, community leaders, religious leaders? Have you analyzed the results by type of participant and were there any differences found? Please state. In the text it is stated that “Salvador focus-groups had higher engagement than those in Jundiai” Explain what you mean with that. Higher in numbers? More engaged in the discussion? And how you were able to tell. 8. Please revisit and improve statements in the intro: a. “Rapid or unplanned 51 urbanisation has contributed to the metropolitan success of this species, which breeds in areas with poor 52 drainage, waste accumulation and open sewers.[10]” The reviewer could not locate this information in reference 10. Besides, ref 10 is about Thailand. Brazil has a long and rich tradition of entomological surveys and the Aedes aegypti and albopictus breeding sites have been well characterized for years. Please cite a Brazilian reference on the breeding sites of Ae ae which are definitely not those mentioned – quite the opposite. Ae ae breeds in man-made containers with walls used for water storage or unintentionally filled with rain-water. Ref 9 is not a good source to look this up either, because it mentions Ae ae only twice in 78 pages. b. “Negotiating responsibility for maintenance of 53 communal spaces or failing to identify cryptic breeding sites prevents adequate vector-control.[11]” These are limitations indeed, but control has not even been achieved in and around the home where breeding abandons and the responsibility does not need negotiation! Responsibility there lies with the household. Yes, there could be a few cryptic breeding sites remaining, underneath vegetation e.g., though it is albopictus with the most cryptic breeding sites, not aegypti. But the majority of breeding sites are non-cryptic at all, exposed out in the open for everyone to see. The disregard for control of immatures and the emphasis on repellents which do not yet exist is unexplained by this reviewer. 9. In the results, please improve clarity: “More participants could describe 138 Ae. aegypti in Salvador, but not all could differentiate the mosquito from other biting insects” There are no questions in the field guide allowing a differentiation – please explain. 10. Ref 9 and any others like it – please provide weblink and date of access. 11. Key words: could add community perceptions, Zika. Medical entomology, Epidemiology listed are off topic.
--	---

REVIEWER	Rajiah, Kingston
REVIEW RETURNED	International Medical University, Department of Pharmacy Practice 12-Aug-2021

GENERAL COMMENTS	Title: As the data has been collected from a single site- a University hospital and two Primary Care Units, it is better to mention it as 'selected region' instead of 'Brazil' in the title. Background: The following statement is not clear. "However, there is no vaccine for ZIKV, and Aedes mosquitoes continue to transmit arboviruses worldwide". Kindly address what is the relationship between WHO declaration and non-availability of vaccines for ZIKV. The following statement should be referenced accordingly.
---

	Currently, one reference is cited. Is that all the literature available? "Despite this, there are few examples of a direct post-epidemic engagement or research co-production with disempowered groups, including pregnant women, communities with lower socioeconomic position and those experiencing racial discrimination". AIM: It would be helpful to provide an explanation about "sociopolitical" in the introduction and why this community knowledge, attitudes and perceptions towards ZIKV is important. Also, there is no connection between the introduction and the aim. It would be better to provide the research gap in the introduction. Methods The "reproductive age (18–49)" is open to more than one interpretation. The authors may consider whether to use this word. The authors could explain why FGD alone was done among women and SSI alone was done among men. The authors could explain why the participants stratified by age (18–30 or 31–49 years) while pregnancy is not taken into account. The authors could add separate subheadings for FGD and SSI and explain the number of FGD conducted and when sample size ended etc under the methods The HBM model appears all of a sudden in the methods section. It would be appropriate to provide information about it under the introduction section. Conclusions: The conclusions should address your research objectives (1) elaborate household preferences for vector-control strategies (2) identify perceived barriers to adoption of preventive behaviours (3) contrast perceptions of ZIKV control over other mosquito-borne arboviruses (4) compare normative beliefs of pregnancy postponement and abortion to reduce foetal susceptibility to CZS (5) map themes against a theoretical framework for behaviour change
--	---

VERSION 1 – AUTHOR RESPONSE

Responses to comments by Reviewer 1

- The field guide provided shows insufficient differentiation among three types of measures: to control the immature vector stages, to control the adult mosquitoes, and to repel their bites. The responses therefore are equally as vague as the question asked. Please acknowledge as a study limitation.**

Response: We would like to thank the reviewer for this comment. This has now been addressed in the first two lines of the limitations section (lines 515 to 518).

However, we note that this question allowed for a broader exploration around the theme of vector control, which being open-ended allowed for respondents to discuss issues of controlling larval and adult mosquitoes if they so wished.

2. **Unclear as to why so much emphasis was given to bite prevention – the hardest of all, for a swift day biter that goes unnoticed. Perhaps the authors can provide some justification and acknowledge this as a limitation.**

Response: Thank you for this comment. With a lack of treatment for Zika and its associated pathologies, measures that can effectively reduce transmission are critical. Such measures include government-led as well as individual/community-based mosquito population management, as well as the use of bite prevention technologies. We do not think that this manuscript overly emphasises an importance of bite prevention, particularly as this formed a key part of the national response to ZIKV.[1] Given that *Aedes* mosquitoes bite during the day, there is a need for measures that can protect people when they are away from the relative security of any interventions they may have at home, be these screens on their doors or efforts to remove larval sites. We have stated the relative lack of familiarity with novel repellents as a limitation to the study.

Unclear as to what the novel repellents were. There is a mention of permethrin treated clothing without any information on whether it was available for purchase and use. Inferred from the participants' responses is that this type of clothing was not available for use at the time of data collection. Participants' answers therefore were in no way based on experience but in speculation. Please provide any additional information on this and add it to the study limitations.

Response: Thank you for your comment. We recognise that the lack of familiarity with novel repellents meant that the responses were somewhat speculative, but this has to be accepted when investigating interest in a new technology. We believe that the responses here, and in our focus groups in Colombia will help guide experimental work on such technologies.[2] At the time novel technologies were being considered and developed under a work package within the ZikaPLAN consortium.[3] For this reason, some interviewers offered further clarification around novel repellents when requested by participants. We have also included that some participants were not familiar with novel repellent fabrics as a study limitation. Please see the amended excerpt from the manuscript included in our response to comment 1 above, lines 515 to 518).

The following excerpts from the study transcripts provide further demonstration of this:

___[Int]: *You know that in some parts of the world you are developing a clothing that already has insecticide and a repellent, what do you think about this kind of use and what would that clothing need for you to wear? For example, if it were a uniform that the company distributed. What do you think?*

___[P1] *Ah! It would be nice if they distributed it to us and our community there, [name], because it would be, it would be cool, because it would be a better way to fight the mosquito...*

Male-Jundiai-SSI 7

5. Personal protection interventions

a) What do you think of personal protection resources / alternatives / practices such as mosquito repellent clothing?

___*Price, the smell that has no collateral effect.*

___*Some clothes already have this, children's pyjamas.*

___[Int 1]: *Pyjamas, but think about pyjamas at night. So, it had to be an outfit.*

___It should be a school uniform.
Jundiai-FGD-2

3. **Figure 2 is unreadable as is. There is no room to explain what each concept means. Please cut out some of the minor themes to be able to explain the major ones better.**

Response: Thank you for bringing this to our attention. We have considered this feedback very carefully, and discussed the different options for alternative display of the themes and their relation to one another.

We believe the inclusion of the minor themes in the figure as important for providing context and positioning of how these themes may contribute to an overall picture of the data. This is particularly as they emerged from a very broad range of topics raised by the discussion guide and focus group participants. Without their inclusion, we are concerned some of the richness and breadth of the data may be overlooked by the audience.

However, we recognise the need to provide a more detailed breakdown of the major themes and their definitions, currently offered in Table 2 and in the supplementary codebook file. As we anticipated this article may be intended for a largely digital audience, the file provided is sufficient size required for display in the 'inline' format for BMJ Open. Should further enlargement be required, the reader can click on the popup link.

Therefore, after careful consideration, we respectfully request that the inclusion of minor themes in the redesigned figure is reconsidered with the digital features offered by the journal in mind (see below **Figure 2**).

Figure 2. Concept map of key, major and minor themes for community knowledge, attitudes and perceptions of Zika virus and vector-control strategies in Salvador and Jundiaí, Brazil. Four

key and 12 major themes were mapped to determine whether they credibly fit constructs for behaviour change outlined in the Health Belief Model.[32,33] The key and major themes are further defined in Table 2 and the code book (Supplementary file).

- 4. Intro, methods and results would benefit from improved writing skills. Senior authors, please pull your weight and bring the writing quality up to that of the discussion section.**

Response: Thank you for providing us the opportunity to strengthen the writing in these sections of the manuscript. We have clarified some of the points raised by both you and the second reviewer in these sections. We have also made careful consideration of how we may improve readability, namely simplifying some wording and breaking down longer sentences.

- 5. There were so many good themes that emerged but they are over-synthesized in a couple of generic headings coming from HBM rather than from the data. Please use more headings that reveal the theme below the heading. The data provide little on core HBM determinants (risk perception) and a lot more on barriers of use. A socioecological model, or a behavioral model might be a better underlying model to use.**

Response: Given the breadth of topics addressed in the topic guide, there was limited scope to address the broader determinants in these models for this data analysis. We feel to have done so would detract from other important themes that emerged during the focus group discussions. For this reason, we have included a limitation that acknowledges our choice of HBM is necessarily limited, but it allows for a deep analysis of the individual level issues that can lead to behaviour change. We have also suggested some additional headings in the discussion. Furthermore, we have acknowledged that there are broader determinants that are not addressed in the scope of this qualitative study (lines 525 to 531).

We have also provided some more context in the background section on the social determinants of health in these settings, to better frame the settings in which this study took place (lines 79 to 90).

- 6. Table 1. Units of analysis are not understandable by the reader – say transcripts instead. Add number of participants for each FG. Mark on the Table the gender for each unit. What criteria were used to put certain participants to the same FG? Why so many FGs? Participants were purposefully sampled – please state with what purpose, i.e., the selection criteria. Do we know anything about their profession or SES? Does Table 1 include health professionals, community leaders, religious leaders? Have you analyzed the results by type of participant and were there any differences found? Please state. In the text it is stated that “Salvador focus-groups had higher engagement than those in Jundiai” Explain what you mean with that. Higher in numbers? More engaged in the discussion? And how you were able to tell.**

Response: We have amended the title table, included participant numbers (where available) and added two rows to indicate the semi-structured interviews were with male participants, and the focus group discussions were with female participants. The labelling of interview units of analysis as transcripts has also been amended for clarity.

The number of focus groups conducted was to aim for saturation, as recommended by qualitative best practices, particularly as some focus groups were smaller (size range 4-7 participants) than the desired 4–8 participants per group.[4,5] Other qualitative researchers have discussed a similar challenge with recruitment in this context.[6]

Additional data on occupation and SES of participants was not provided for this data analysis and has been noted in the limitations (lines 525 to 527):

“Additional data were not collected on participants, such as data on socioeconomic position, which along with missing data on age for some Jundiaí focus groups could have provided an interesting overview of the participants in this study.”

To better explain the reasoning for our statement around focus engagement, this was based on the length and depth of discussion observed in transcripts in Salvador by word count. In addition to this observation during analysis, balance of participant contribution to each focus group was crudely quantified for Salvador focus groups in NVivo (version 12, QSR International). Of the 43 women enrolled in Salvador, only 6 participants were noted to have spoken less (highlighted in **Figure 3**), suggesting a relatively balanced level of contribution across the groups as a whole. However, as transcripts were missing data on the breakdown of the 60 women who participated in Jundiaí, we have removed this statement. Figure 3 was omitted from the final analysis due to its limited value in supporting us meet the objectives of this study and its methodological limitations as a comparative proxy for engagement.

Figure 3. Number of references and coverage (%) of each focus group participant in Salvador using NVivo (version 12, QSR International).

FGD-1			FGD-2			FGD-3			FGD-4						
	Ref.	%		Ref.	%		Ref.	%		Ref.	%				
	Ka.	313	34.6		M1	35	16.0		A.	370	51.1		G.	118	23.1
	Ke.	115	13.8		M2	44	37.8		T.	147	20.8		M.	121	20.2
	R.	5	0.2		M3	25	7.4		V.	90	10.0		A.	84	15.6
	A.	465	26.1		M4	10	4.5		G.	2	0.0		B.	152	39.5
	D.	175	22.7		M5	37	11.6		L.	116	17.0				
	K.	7	0.6		M6	38	16.1								

FGD-5			FGD-6			FGD-7			FGD-8						
	Ref.	%		Ref.	%		Ref.	%		Ref.	%				
	M1	42	12.0		C.	313	31.6		M1	129	25.4		M1	80	28.1
	M2	73	20.3		V.	83	6.4		M2	38	2.9		M2	98	36.1
	M3	64	24.2		L.	246	23.3		M3	146	25.3		M3	53	20.5
	M4	72	34.1		J.	236	27.3		M4	112	13.6		M4	41	13.6
	M5	30	7.3		M.L.	46	6.6		M5	197	26.4		M5	1	0.1
					M.J.	44	4.0		M6	16	1.4				
									M7	10	1.4				

7. Please revisit and improve statements in the intro:

- a. **“Rapid or unplanned 51 urbanisation has contributed to the metropolitan success of this species, which breeds in areas with poor 52 drainage, waste accumulation and open sewers.[10]”** The reviewer could not locate this information in reference 10. Besides, ref 10 is about Thailand. Brazil has a long and rich tradition of entomological surveys and the *Aedes aegypti* and *albopictus* breeding sites have been well characterized for years. Please cite a Brazilian reference on the breeding sites of *Ae ae* which are definitely not those mentioned – quite the opposite. *Ae ae* breeds in man-made containers with walls used for water storage or unintentionally filled with rain-water. Ref 9 is not a good source to look this up either, because it mentions *Ae ae* only twice in 78 pages.

Response: We would like to thank the reviewer for this comment and have included suggested alternatives for references for this statement, as well as amended the sentence (lines 51 to 54), please see in-text amendments below in response to 7.b.

- b. **“Negotiating responsibility for maintenance of 53 communal spaces or failing to identify cryptic breeding sites prevents adequate vector-control.[11]”** These are limitations indeed, but control has not even been achieved in and around the home where breeding abandons and the responsibility does not need negotiation! Responsibility there lies with the household. Yes, there could be a few cryptic breeding sites remaining, underneath vegetation e.g., though it is albopictus with the most cryptic breeding sites, not aegypti. But the majority of breeding sites are non-cryptic at all, exposed out in the open for everyone to see. The disregard for control of immatures and the emphasis on repellents which do not yet exist is unexplained by this reviewer.

Response: We do not feel that we have disregarded the control of immature mosquitoes, and indeed have highlighted that the most frequent vector-control strategy described by participants at the household level was preventing water stagnating by recycling, using sand and covering open receptacles (lines 318 to 320). However, we found it of interest to also investigate more novel approaches that may be accepted by communities and which are relatively unexplored; if larval sites are found and treated/removed around the house, there is still a need to protect people when they leave their households.

In addition, drawing from the data, we feel that the responsibility for vector control does indeed require negotiation. In particular, several participants from both Salvador and Jundiaí focus groups discussed in depth the difficulty determining who in the periodomestic was the responsible party for maintenance of these spaces, whether it was the individual, their neighbours, municipal service providers, or whether responsibility lay further upstream. We have therefore revised this section of the manuscript to clarify what we feel might be a misinterpretation of what we intended by these statements (lines 51 to 57). We would also like to present a selection of excerpts from the data to further evidence the analysis of these participant-led discussions:

___[M]: *Another thing, another question, also that we end up putting after the outbreak. As we always know, we take care of our house, but the neighbour does not.*

Salvador-FGD8

I have a neighbour at home, she doesn't even care. I took it myself and went to clean her backyard. Because I'm afraid, my son has already been hospitalized because of that. She has three young children, she doesn't care to clean the yard.

Jundiaí-FGD-9

8. **In the results, please improve clarity: “More participants could describe 138 Ae. aegypti in Salvador, but not all could differentiate the mosquito from other biting insects” There are no questions in the field guide allowing a differentiation – please explain.**

Response: The topic guide was designed to omit detail of the physical description of *Ae. aegypti* or its distinction from other biting insects. This was to ensure questions were more top line and did not bias participant responses. Questions were therefore not specific to *Ae. aegypti* as they aimed to capture the present depth and accuracy of participant knowledge of Zika virus and other mosquito-borne disease competent vectors.

During the analysis, it became apparent that a detailed description of the mosquito's appearance and behaviours were sometimes raised during the Salvador focus groups, whereas this was not discussed by Jundiaí participants. However, most participants from both sites did not differentiate

between vectors, and several indicated misunderstanding around identification of ZIKV-competent vectors. For example:

___[P2]: *There are 3 different mosquitoes, right?*

Salvador-FGD-2

___[P1]: *I'll say, the red bugs, I do not know if it's muriçoca or if it's Zika, understand? ...then I wonder: "Is it muriçoca or is it a Dengue mosquito? Underst[and]? We take care, but you never know, right? At least differentiate the mosquito from Dengue, I do not know.*

Salvador-FGD-6

To improve clarity in this statement, we have amended as follows and included the first excerpt above from Salvador-FGD2 in the results section (lines 166 to 167).

9. Ref 9 and any others like it – please provide weblink and date of access.

Response: As we have included additional references in response to review comments 7.a., we have therefore taken this opportunity to instead reference Singh *et al.* (2018) in place of the original reference 9.[7] We have also made a thorough check of the bibliography and included access date and a weblink for all references which required this.

10. Keywords: could add community perceptions, Zika. Medical entomology, Epidemiology listed are off topic.

Response: Thank you for suggesting alternative keywords for our manuscript. We attempted to include the below as suggested:

Qualitative study, Community perceptions, vector control, Zika virus, Arboviruses

However, as the journal portal only permits key words to be selected from a drop down menu we had limited control over this. We have therefore selected the following from the options provided:

Subject headings: Public health, Qualitative research, Infectious diseases, Global Health, Health policy

Key words: Public health, Infection control, Epidemiology, Entomology

Responses to comments by Reviewer 2

1. As the data has been collected from a single site- a University hospital and two Primary Care Units, it is better to mention it as 'selected region' instead of 'Brazil' in the title.

Response: Thank you for this comment. We have made it more explicit that data collection took place across 4 different sites in two distinct regions of Brazil: Bahia state in the north east and São Paulo state in the south east of Brazil (lines 535 to 537):

Additional information on the study setting is provided in the background section (lines 94 to 96) and in the Patient and Participant Involvement section (lines 22 to 25).

2. **The following statement is not clear:**
"However, there is no vaccine for ZIKV, and Aedes mosquitoes continue to transmit arboviruses worldwide". Kindly address what is the relationship between WHO declaration and non-availability of vaccines for ZIKV.

Response: We would like to thank you for suggesting on how we can strengthen the clarity of this statement. We have revised the wording and included additional supporting text with a reference to provide further context (lines 74 to 79).

The following statement should be referenced accordingly. Currently, one reference is cited. Is that all the literature available?" *Despite this, there are few examples of a direct post-epidemic engagement or research co-production with disempowered groups, including pregnant women, communities with lower socioeconomic position and those experiencing racial discrimination*".

Response: Many thanks for this comment. We have since added additional relevant references and amended the wording for this section for clarification (lines 83 to 90).

3. ***It would be helpful to provide an explanation about "sociopolitical" in the introduction and why this community knowledge, attitudes and perceptions towards ZIKV is important.***

Response: In joint response to reviewer 1 and your question 3, we have provided some additional context on the social determinants of health in the background section (lines 54 to 56 and 74 to 92), which also follows on to the study aims detailed in question 5 below.

4. **Also, there is no connection between the introduction and the aim. It would be better to provide the research gap in the introduction.**

Response: We would like to thank reviewer 2 for this comment and have edited the introduction (lines 83 to 90), and aims section to better clarify the gap, need and aims of this study (lines 95 to 104).

5. **The "reproductive age (18–49)" is open to more than one interpretation. The authors may consider whether to use this word.**

Response: Thank you for bringing this to our attention. As suggested, we have amended our wording in the abstract and methods sections to clarify that adults of approximate reproductive age were included in this study. These were stratified into two 30-year age ranges, which we feel sufficient for the objectives of this study (18–30 and 31–49 years). Please also see our response to question 8 below.

6. **The authors could explain why FGD alone was done among women and SSI alone was done among men.**

Response: The focus of this investigation was on messaging that targets pregnant women and other women of reproductive age, which was the study's perceived most at-risk group, to contribute to the prevention of CZS cases/sexual transmission. Whilst partners of these women were important due to their intimate relationship with the pregnant women and other women of reproductive age in this study, since they were not the main focus of the study, FGDs were not conducted with this group as well.

- 7. The authors could explain why the participants stratified by age (18–30 or 31–49 years) while pregnancy is not taken into account.**

Response: Thank you for this comment. Firstly, we wanted to examine at risk populations, for example, those most at risk of adverse outcomes following in utero Zika infection. We also wanted to stratify into younger and older groups and chose 30 years as the (arbitrary but reasonable) cut-off to separate the two, with the range of the study objectives and participant sample size in mind.

- 8. The authors could add separate subheadings for FGD and SSI and explain the number of FGD conducted and when sample size ended etc under the methods.**

Response: Thank you for this suggestion. We have provided additional information in Table 1 to clarify numbers participating in focus groups and semi-structured interviews (line 141). However, to be mindful of the manuscript word count due to our revisions in responses to other questions, we have not discussed sampling in more depth in the methods section. However, more context on our approach to recruitment through community stakeholders has been provided in the Patient and Public Involvement section (lines 123 to 130). Sampling is also discussed in the 'Strengths and limitations of this study' box before the abstract.

- 9. The HBM model appears all of a sudden in the methods section. It would be appropriate to provide information about it under the introduction section.**

Response: Many thanks to the reviewer for their comment regarding the positioning of the introduction to the HBM in the manuscript. We have since included the following text to the introduction (lines 89 to 92) and aims (lines 95 to 99) sections, respectively.

- 11. The conclusions should address your research objectives**

- (1) elaborate household preferences for vector-control strategies**
- (2) identify perceived barriers to adoption of preventive behaviours**
- (3) contrast perceptions of ZIKV control over other mosquito-borne arboviruses**
- (4) compare normative beliefs of pregnancy postponement and abortion to reduce foetal susceptibility to CZS**
- (5) map themes against a theoretical framework for behaviour change**

Response: Thank you for offering us the opportunity to elaborate further on how findings met our study objectives in the conclusion (lines 560 to 572). We have taken care to address each objective whilst structuring the conclusion in a way that raises each theme in a succinct yet relevant order following on from the discussion.

References

- 1 Ministry of Health (BR). Plano nacional de enfrentamento à microcefalia: Protocolo de atenção à saúde e resposta à ocorrência de microcefalia (v3). Brazil: Ministério da Saúde 2016. https://bvsmms.saude.gov.br/bvs/publicacoes/protocolo_atencao_saude_resposta_ocorrencia_micr_cefalia.pdf [Accessed 08 February 2021]
- 2 Mendoza C, Jaramillo GI, Ant TH, et al. An investigation into the knowledge, perceptions and role of personal protective technologies in Zika prevention in Colombia. *PLoS Negl Trop Dis* 2020;14(1):e0007970. doi: 10.1371/journal.pntd.0007970 [published Online First: 2020/01/22]
- 3 Wilder-Smith A, Preet R, Brickley EB, et al. ZikaPLAN: addressing the knowledge gaps and working towards a research preparedness network in the Americas. *Global Health Action* 2019;12(1):1666566. doi: 10.1080/16549716.2019.1666566
- 4 Braun, V, Clarke V. Using thematic analysis in psychology. *Qual Res Psychol* 2006;3:77–101. doi:10.1191/1478088706qp063oa
- 5 Krueger RA, Casey MA. Focus groups: A practical guide for applied research (5th Ed). California: SAGE Publications 2015.
- 6 Passos MJ, Matta G, Lyra TM, et al. The promise and pitfalls of social science research in an emergency: lessons from studying the Zika epidemic in Brazil, 2015–2016. *BMJ Global Health* 2020;5(4):e002307. doi: 10.1136/bmjgh-2020-002307
- 7 Singh RK, Dhama K, Khandia R, et al. Prevention and control strategies to counter Zika virus, a special focus on intervention approaches against vector mosquitoes—Current Updates. *Frontiers in Microbiology* 2018;9(87) doi: 10.3389/fmicb.2018.00087